# Proteomics Analysis of Urine to Examine Physical Effects of Warm Nano Mist Sauna Bathing

**DOI:** 10.3390/healthcare7020071

**Published:** 2019-05-19

**Authors:** Yoshitoshi Hirao, Naohiko Kinoshita, Bo Xu, Suguru Saito, Ali F. Quadery, Amr Elguoshy, Keiko Yamamoto, Tadashi Yamamoto

**Affiliations:** 1Biofluid Biomarker Center, Niigata University, Niigata 950-2181, Japan; hirao@ccr.niigata-u.ac.jp (Y.H.); kinoshita@nuhw.ac.jp (N.K.); kyonami-bbc@ccr.niigata-u.ac.jp (B.X.); scn.nth@gmail.com (S.S.); fquadery@yahoo.com (A.F.Q.); amr_biotech2006@yahoo.com (A.E.); yamamotok-bbc@ccr.niigata-u.ac.jp (K.Y.); 2Department of Health Informatics, Faculty of Healthcare Management, Niigata University of Health and Welfare, Niigata 950-3198, Japan; 3Department of Clinical Laboratory, Shinrakuen Hospital, Niigata 950-2087, Japan

**Keywords:** sauna bathing, health assessment, urine

## Abstract

Conventional sauna bathing may have some health benefits as it facilitates relaxing, detoxing and promoting metabolism. However, conventional sauna bathing at a high temperature may be harmful for the body by increasing the risk of heart failure. The nano-mist sauna has been developed to provide nano-size water particles at a lower temperature. Hence, nano-mist sauna bathing is expected to be useful for health without the risks that arise at high temperatures. In this study, we performed a comprehensive proteomics analysis of urine samples obtained from healthy volunteers before and after they had taken a sauna bath with nano-mist (*n* = 10) or with conventional mist (*n* = 10) daily for two weeks (4 groups). The average numbers of urine proteins identified by liquid chromatography-linked mass chromatography in each group varied from 678 to 753. Interestingly, the protein number was increased after sauna bathing both with nano-mist or with conventional mist. Quantitative analysis indicated that considerable numbers of proteins were obviously up-regulated, with more than two-folds in urine samples after the nano-mist sauna bathing. The KEGG pathway analysis showed significant stimulation of the lysosome pathway (*p* = 5.89E−6) after the nano-mist bathing, which may indicate the nano-mist sauna bathing promotes metabolism related to the lysosome pathway more efficiently than conventional mist sauna bathing and may provide more health benefits.

## 1. Introduction

Urine remains as one of the preferred biological samples for diagnosis and disease monitoring, due to its non-invasive nature of collection and to its simpler matrix as compared to other blood derived fluids [1]. Urine collection is simple, non-invasive, and available volume is relatively abundant compared to other biological fluids. It contains information from systemic circulation to local tissues via extracellular vesicles, proteins and small molecules. However, despite its lower range of protein concentrations, highly abundant proteins such as albumin, representing 25% of the total protein amount, are still a hurdle to the complete characterization of the urinary proteome, as they may mask the least abundant proteins [2]. Differential protein profiles and abundances are often observed for many diseases from proteomics studies [3]. The standardized definition of proteomics biomarkers was proposed as “a specific peptide or protein that is associated with a specific condition, such as the onset, manifestation, or progression of a disease or a response to treatment” [4]. 

Recently, sauna bathing has widely spread worldwide from Nordic countries. It has been reported that regular sauna bathing can deep clean the skin, promote weight loss, improve blood circulation, accelerate muscle recovery, relieve tension headaches, and induce deeper and more relaxing sleep [5,6,7,8]. In addition to the traditional dry sauna bathing that heats the sauna room temperature to 70–100 °C, mist sauna bathing that sparges hot water into the sauna room and maintains the room temperature at 40–45 °C has also become prevalent. Many investigations into dry sauna bathing have been conducted due to its medical availability; however, few reports have mentioned that dry sauna bathing has higher risks for the heart than mist sauna bathing [9,10,11]. Traditional dry sauna bathing has many benefits, but also has disadvantages due to the risk of death caused by drugs [9]. Since the beneficial effects of sauna bathing described above are considered to be due to the results of sweating, the same effects might be obtained through mist sauna bathing. The suppression of evaporative function in mist sauna bathing might comprise highly effective heating and the prevention of dehydration [12].

In this study, we utilized a nano mist sauna, which is a new type of sauna characterized by the ability to produce nano sized mist (Figure 1). This type of sauna has some benefit such as a low temperature, high humidity and being highly effective for triggering sweating. Here we compared comprehensive urine proteomes with and without a nano mist sauna to describe the profile of the change in the human internal pathway.

## 2. Materials and Methods 

### 2.1. Sauna Bathing and Human-Derived Urine Sample Collection

Urine samples were obtained from healthy male individuals. Twenty healthy volunteers were recruited; and 10 volunteers bathed in a nano mist sauna for 20 min, while the other 10 bathed in a non-nano mist sauna for 20 min. Urine sample collection was performed twice for each group. On the first day, urine was collected prior to starting the study. A second urine sample was taken following 2 weeks of daily sauna use. Collected urine was immediately centrifuged by 1000× *g* for 15 min. Then, the removed supernatant was moved to a new collection tube for avoid cell debris. The supernatants were separated as aliquots in 1.5 ml tubes and stored at −20 °C until use.

### 2.2. Urine Protein Preparation

Frozen urines were thawed in a water bath at 37 °C for 10 min before precipitation. The urine proteins were precipitated by Methanol/chloroform precipitation from 500 µL of urine. An equal volume of methanol and one forth of the chloroform were added to the urine sample, then mixed well for 5 min. The sample was centrifuged at 19,000× *g* at 25°C for 15 min. The supernatant was discarded without interfering with the interface layer using a pipette. Then, an equal volume of methanol was added to the sample again and mixed gently for 5 min. The sample was centrifuged at 19,000× *g* at 25 °C for 15 min. Finally, the supernatant was removed and obtained proteins were dried by air.

### 2.3. Urine Peptide Preparation

Precipitated proteins were dissolved in 100 µL of 8 M urea, with a 50 mM Tris-HCl (pH 8.0) buffer. The sample was treated by 2 µL of 1 M Dithiothreitol at RT for 1 h and 8 µl of 500 mM Iodoacetamide at RT for 1 h with shading. The alkylation was stopped by 1 µL of 1 M Dithiothreitol and then diluted eight times by 50 mM Tris-HCl (pH 8.0). For the digestion, 1 µg of trypsin (Agilent, Santa Clara, CA, USA) was added to the sample and incubated at 37 °C for 16 h with shaking. The digestion was stopped by the addition of 1 µL of 50% Trifluoro acetic acid (TFA). The digested sample was purified by C18 spin column (GL Science, Tokyo, Japan) according to the manual. Briefly, a C18 spin column was activated by 100% and 50% acetonitrile sequentially and then equilibrated by 0.2% TFA with centrifuging at 3000× *g* for 30 sec. After conditioning, the sample was loaded into a spin column and centrifuged at 3000× *g* for 90 sec. Then, trapped peptides were washed by 0.2% TFA twice and eluted by 95% acetonitrile with 5% formic acid. The eluted sample was dried up by a VEC-260 vacuum dryer (Iwaki, Tokyo, Japan). The sample was re-suspended by 0.1% formic acid and the peptide concentration was measured by Nano drop 1000 (Thermo, Bremen, Germany). The sample was stored at −80 °C until use.

### 2.4. LC-MS/MS Analysis and Protein Identification

Samples were analyzed on a QExactive plus (Thermo) in data dependent acquisition. Peptides were resolved by nanoflow liquid chromatography system (easy nLC1000, Thermo) on a trap column (2 cm × 75 µm Acclaim Pepmap 100 column) and a separation column (12.5 cm × 75 µm NTCC-360) at a 300 nL/min with a multistep gradient. Mobile phase A: water with 0.1% formic acid; mobile phase B: acetonitrile with 0.1% formic acid. The 500 ng of peptides injected and eluted from the analytical column with in a linear gradient 2%B to 35%B in 120 min. The mass spectra were obtained in a positive ion mode in the scan range MS and MS/MS of 350–1800 m/z and 200–2000 m/z, respectively. The 15 most intense peaks with charge state 2 and 3 were selected from each survey scan and subjected to CID fragmentation. For the MS and MS/MS, scan parameter settings are as follows: collision energy, 35%; electrospray voltage, 2.0 kV; capillary temperature, 250 °C and isolation windows 4 m/z. The raw files acquired by mass spectrometry were transferred to mgf files by MSConvert [13]. Then, proteins and peptides were identified via MASCOT search (v2.3.1, Matrix Science, Boston, MA, USA) with database searching. The data were queried against a Unipro/SWISS-prot database (v2015-08; Homo sapiens 20,203 sequences). All database searches were performed using a precursor mass tolerance of ± 10 ppm, a fragment ion mass tolerance of ± 0.02 Da, enzyme specificity was set to trypsin and maximum missed cleavage values of 2. Cysteine carbamidomethylation was set as a fixed modification. The false discovery rate (FDR) to 1% on peptide spectrum match level. For label free semi-quantification, we handled a Normalized Spectral Index (SI_N_) in this study [14]. The datasets were compared by Venny [15] in this study. Gene Ontology (GO) analysis was performed by the PANTHER (ver. 13.1) [16,17] and Kyoto Encyclopedia of Genes and Genomes (KEGG) pathway analysis was analyzed by DAVID (ver. 6.7). All the results of GO analysis and KEGG pathway analysis were treated by Microsoft EXCEL software. All MS files (.raw) and MASCOT result files were accessible from the JPOST repository [18]. 

## 3. Results and Discussion

To achieve a comprehensive analysis of the urinary proteins by LC-MS/MS, a duplicate analysis of the same sample was performed for urine samples. The amount of each protein was semi-quantitated by the SI_N_ value and compared statistically between the two groups. The proteins up- or down-regulated in the sample after bathing in a nano mist sauna were analyzed by the GO annotation and KEGG pathway tools to elucidate the characteristics of the proteins and pathways (Figure 2).

### 3.1. Protein Identification and GO Annotation

For the nano mist sauna group, the average number of proteins identified in the before and after were 678 and 840, respectively (Figure 3). From the Venn diagram, 64 unique proteins for the before group, 229 unique proteins for the after group and 774 shared proteins were demonstrated. On the other hand, for the group without nano mist sauna bathing, 753 and 788 proteins were identified before and after bathing in saunas, respectively. From this result, only 4.6% increased after bathing in a sauna without nano mist. By contrast, the number of identified proteins were increased by 24% in the nano mist sauna group (Figure 3B). This result indicates that the nano mist sauna affected and changed protein expressions in the human body.

Furthermore, each group of proteins was demonstrated to distribute for Biological Process by a Gene Ontology analysis (Figure 4). The top 3 results of a BP analysis were cell adhesion, biological adhesion and the response to wounding for all groups. From GO annotation, most of the proteins which were identified in each group were not significantly different.

### 3.2. Label Free Semi-Quantification and KEGG Pathway Analysis

From protein identification, we performed label free semi-quantification by SI_N_ from each signal intensities for all identified proteins. Then, each SI_N_ were compared between the groups and the proteins were extracted, which were identified more than seven samples in each group. These extracted proteins of SI_N_ were compared between before and after taking a sauna with/without nano mist. Subsequently, we selected the proteins, which were increased or decreased more than two times in the groups with and without exposure to a nano mist sauna. Finally, we performed KEGG pathway analysis by DAVID using changed proteins (Table 1). It was clear from Table 1 that several pathways were dramatically changed and their details were different in each group. For instance, with the nano mist sauna group, the highly affected pathway was the Lysosome pathway. The lysosome pathway works through synthesis and digestion and can lead to endocytosis, phagocytosis and autophagy. Furthermore, Gowrishankar reported that the lysosome pathway can affect Alzheimer’s disease [19]. These results suggested that bathing in a nano mist sauna affected the body and enhanced synthesis and digestion. Table 2 indicates a list of proteins which were unique or highly increased at the group that bathed in a nano mist sauna. Interestingly, Arylsulfatase B, identified in the group that bathed in a nano mist sauna, has a key role in metabolism [20]. In addition, we picked up a few proteins which were related to lysosome pathways. Lysosomal thioesterase PPT2 (PPT2) was found in our list and this protein can digest the palmitic acid in the lysosome pathway [21,22,23]. This report also supports our proteomics data. Hence, our result is thought to be the nano mist sauna effect and enhances the internal body metabolism.

## 4. Conclusions

We performed comprehensive analysis using human urine to determine the effects of bathing in a nano mist sauna for healthy volunteers. This study is one of the first reports to describe the human health care with MS-based proteomics approach. Sauna bathing has a therapeutic value. Some studies have suggested that regular sauna bathing may lower the blood pressure in patients with hypertension and increase the left ventricular ejection fraction in patients with chronic heart failure [9]. However, sauna bathing also has the risk of causing coronary heart disease. Nano mist sauna has some benefits due to its temperature, humidity and so on. For example, the temperature of the nano mist sauna increased to a maximum of 40°C. This temperature is milder than for the dry sauna (70–100°C) and also does not have a risk of causing heart damage. Furthermore, the humidity of nano mist sauna reaches almost 100%. Despite the high humidity, bathing under these circumstances is not painful due to the low temperature. This may be useful for enhancing human metabolism.

With the wide application of the urinary proteome in healthcare research, the construction of a representative and informative normal urinary database has become critically important. In this study, we reported the comprehensive analysis of a urinary proteome to profile the effects of nano mist sauna bathing. Our findings can be helpful for discovering the biomarker or the index for health checking in the future.

## Figures and Tables

**Figure 1 healthcare-07-00071-f001:**
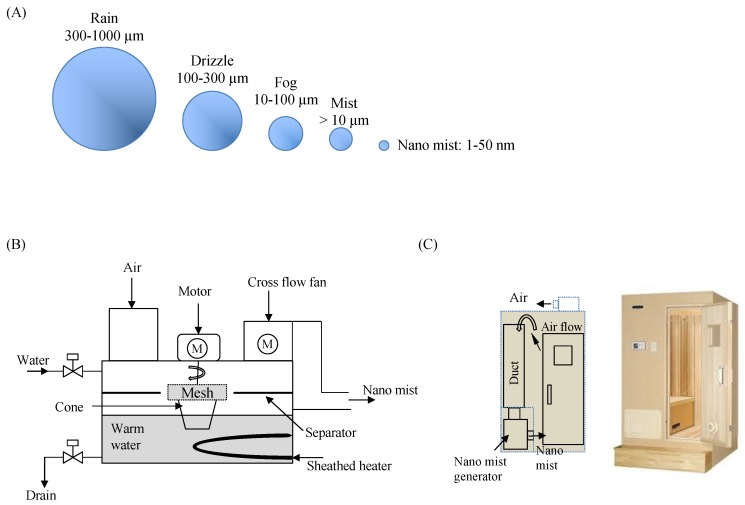
(**A**) Each size of water particles compared with nano mist. (**B**) System diagram of nano mist generation. (**C**) Outline of nano mist sauna.

**Figure 2 healthcare-07-00071-f002:**
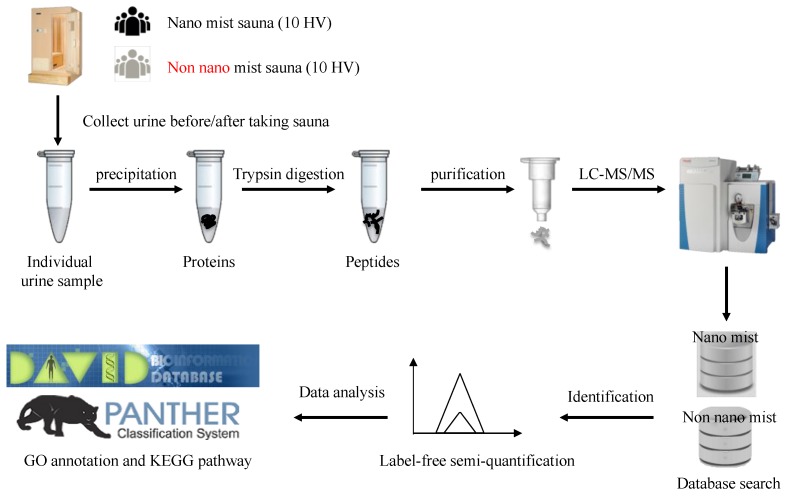
Schematic illustration of the urine proteomic screening process.

**Figure 3 healthcare-07-00071-f003:**
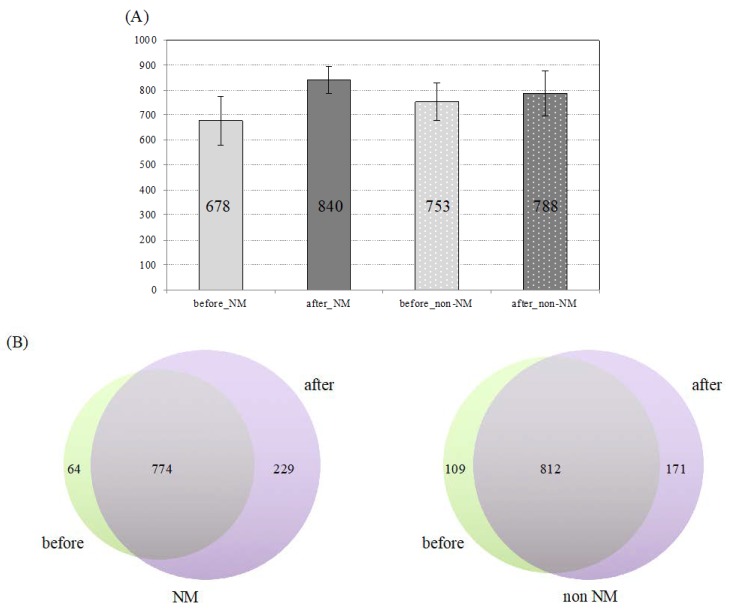
Protein identification of each group. (**A**) Average protein identification number with the non redundant results from each groups with a < 0.05 significant threshold. The error bar indicates standard deviation. (**B**) Comparison of unique and shared proteins both before and after bathing in a sauna. The left Venn diagram indicates a nano mist sauna and right panel shows a non nano mist sauna (detailed information can be found in the Appendix A).

**Figure 4 healthcare-07-00071-f004:**
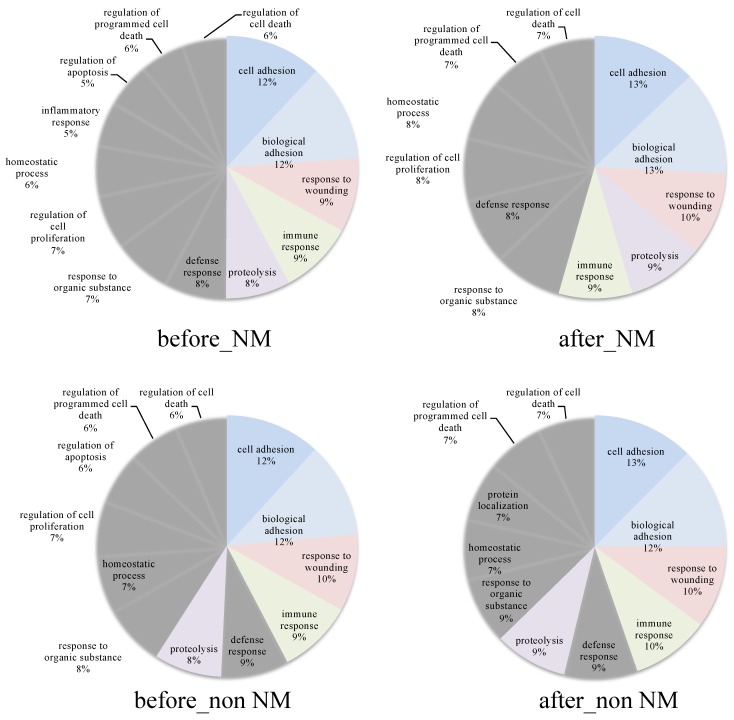
Circle graph showing biological processing from GO annotation. The top 5 biological processes were colored.

**Table 1 healthcare-07-00071-t001:** List of changed pathway in each group by bathing in a sauna.

Pathway	Count	*p* Value
(A) nano mist, before to after > 2.0 including unique value for after 302 proteins		
Lysosome	15	5.89E-6
Other glycan degradation	6	6.58E-5
ECM-receptor interaction	11	1.4E-4
Pyruvate metabolism	7	9.31E-4
(B) nano mist, before to after > 0.5 including unique value for before: 125 proteins		
not detected	-	-
(C) non nano mist, before to after > 2.0 including unique value for after: 216 proteins		
Glycolysis / Gluconeogenesis	10	1.06E−6
Pyruvate metabolism	6	7.29E−5
Pathogenic Escherichia coli infection	6	3.63E−4
(D) non nano mist, before to after > 0.5 including unique value for before: 234 proteins		
Axon guidance	9	2.15E−6
Focal adhesion	11	3.16E−5
Prion disease	5	3.42E−4
ECM-receptor interaction	7	3.89E−3

**Table 2 healthcare-07-00071-t002:** The identified proteins which related to the lysosome pathway at more than 2 times fold changed.

Accession No.	Description	Fold Change
P15848	Arylsulfatase B	unique
P17900	Ganglioside GM2 activator	2.48
P06865	Beta-hexosaminidase subunit alpha	2.37
P07686	Beta-hexosaminidase subunit beta	2.67
P10619	Lysosomal protective protein	2.55
P08962	CD63 antigen	unique
O00462	Beta-mannosidase	unique
P16278	Beta-galactosidase	2.61
O00115	Deoxyribonuclease-2-alpha	unique
P06280	Alpha-galactosidase A	3.06
P34059	N-acetylgalactosamine-6-sulfatase	2.21
Q99523	Sortilin	unique
Q14108	Lysosome membrane protein 2	unique
Q9HBG4	V-type protein ATPase 116 kDa subunit a isoform4	unique
P04062	Glucosylceramidase	unique

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
