# Peer review of "Proteomics Analysis of Urine to Examine Physical Effects of Warm Nano Mist Sauna Bathing"

_healthcare, 2019, doi:10.3390/healthcare7020071_

Reviewer 1 Report

Hirao et al have carried out an interesting study which is becoming quite relevant in some fields and certainly deserves attention. This manuscript is well written although several typos are present. There are two concerns with this manuscript, the first and most important is that there appears to be no control for variation in the urine concentration before and after the sauna. Clearly after the sauna the participants will be more dehydrated than before thus producing a more concentrated urine sample. This must be considered before the work can be published as it’s likely to significantly skew the results. There are several good papers on normalising urine concentration (mainly from the metabolomics community). The second issue is that he reporting of protein identities does not conform to MIAPE guidelines and should be corrected possibly as a supplementary information.

Introduction

Page 1 line 35: Remove “T”

Page 2 line 54: Is the risk of death by drug and alcohol due to taking them at the time of sauna?

Page 2 line 61: correct “effectd”

Materials and methods

Page 2 and 3, lines 69-74: How long before and after the sauna were the samples taken? Clearly there is bias here and any urine produced after the sauna will be more concentrated due to dehydration.

Page 3, lines 76-77: Shouldn’t the frozen urine sentence come before the precipitation?

Page 3 line 88: “stopped by 50% TFA” does this mean the TFA was added to make solution up to 50% or a set volume of 50% TFA was added?

Page 3 line 103: change “mass spectrometer was obtained in”, to “the mass spectra was obtain in”

Page 3, lime 114: “we” everything has been in 3rd person up to here.

Results and discussion

Page 4, lines 130-137: IT cannot be stated that the saunas changed human protein expression levels as there was no normalisation for the urine concentration which is bound to have changed through dehydration during the sauna.

The reporting of identified proteins to not conform with MIAPE guidelines and should be altered to do so: http://psidev.info/sites/default/files/2018-03/MIAPE_MSI_1.1.pdf

Author Response

Dear Reviewer 1

              We would like to thank you for the valuable comments concerning our manuscript. We believe these suggestion/criticisms have helped us to improve our manuscript. We are glad to learn of the reviewer’s opinion that our manuscript may be suitable for publication in Healthcare after some revision. Please find attached a revised version of our manuscript entitled “Proteomics Analysis of Urine to Examine Physical Effects of Warm Nano Mist Sauna Bathing” by Hirao et al. (healthcare-496639). The manuscript has been revised in accordance with the reviewer’s comment and all the issues raised have been addressed. In Changes to the manuscript are highlighted in red and one of the table is added.

              We greatly appreciate the careful assessment of our manuscript by yourself and the reviewers. We hope the manuscript is now acceptable for publication in Healthcare.

Yours sincerely,

Tadashi Yamamoto, M.D & Ph.D.

Response to the comments

Comment-1

              Page 1 line 35: Remove “T”

Response-1

              We removed “T” from manuscript.

Comment-2

              Page 2 line 54: Is the risk of death by drug and alcohol due to taking them at the time of sauna?

Response-2

              The sentence we prepared was wrong. We corrected it to “Traditional dry sauna bathing has many benefits but also has disadvantages for taking due to have a risk of death.” at Page 2 line 53-55.

Comment-3

              Page 2 line 61: correct “effectd”

Response-3

              We corrected “effectd” to “effected”.

Comment-4

              Page 2 and 3, lines 69-74: How long before and after the sauna were the samples taken? Clearly there is bias here and any urine produced after the sauna will be more concentrated due to dehydration.

Response-4

              For the sample collection, there was a lack of explanation. The schedule of collecting urine is as follow; At 1st day, urine was collected for before taking sauna. Then volunteer took sauna every day for 2 weeks. At end of 14th days, we again collected the urine for after taking sauna. From your suggestion, we corrected the sentence “At 1st day, urine was collected for before taking sauna. Then volunteer took sauna every day for 2 weeks. At end of 14th days, we again collected the urine for after taking sauna.” at Page 2 lines 70-72.

Comment-5

              Page 3, lines 76-77: Shouldn’t the frozen urine sentence come before the precipitation?

Response-5

              We collected the sentence to “Frozen urines were thawed in water bath at 37°C for 10 min before precipitation. The urine proteins were precipitated by Methanol/chloroform precipitation from 500 µl of urine.” at Page 3 lines 76-77.

Comment-6

              Page 3 line 88: “stopped by 50% TFA” does this mean the TFA was added to make solution up to 50% or a set volume of 50% TFA was added?

Response-6

              50% TFA was added. We corrected the sentence to “The digestion was stopped by 1 µl of 50% Trifluoro acetic acid (TFA)” at Page 3 lines 88-89.

Comment-7

              Page 3 line 103: change “mass spectrometer was obtained in”, to “the mass spectra was obtain in”

Response-7

              We corrected to “the mass spectra was obtain in” at Page 3 lines 103.

Comment-8

              Page 3, lime 114: “we” everything has been in 3rd person up to here.

Response-8

              We corrected the sentence to “The false discovery rate (FDR)” at Page 3 lines 114.

Comment-9

              Page 4, lines 130-137: IT cannot be stated that the saunas changed human protein expression levels as there was no normalisation for the urine concentration which is bound to have changed through dehydration during the sauna.

Response-9

              This comment is related to comment 4. The urine samples were compared between before and after taking sauna 2 weeks. From this term of taking sauna, we thought that we can profile the effect of nano mist sauna bathing.

Comment-10

              The reporting of identified proteins to not conform with MIAPE guidelines and should be altered to do so: http://psidev.info/sites/default/files/2018-03/MIAPE_MSI_1.1.pdf.

Response-10

              Thank you for your suggestion. We checked the MIAPE guidelines. From the guidelines, we corrected “Table 2” of “Gene name” to “Accession No.” and all the list proteins were changed to use accession No.

Reviewer 2 Report

In the introduction at the beginning of the first line there is a repeated letter.the non-nano mist sauna is sauna or is it sauna wet but not with nano particles?this is a case study, because the sample is very small, and because it was only made in men? women have no benefits, or is there any other reason?How many times have you had a sauna for how long? The urine samples were collected before the test was started or were done every time before doing sauna ... if yes, was it done more than once !?table 1 is not explicit and should be improved so as to be clearer The conclusion relates little to the results obtained with the benefits of the sauna, to say that the nano sauna has benefits because of humidity, temperature etc. without explanation seems to me quite vague and poor.

Author Response

Dear Reviewer 2

              We would like to thank you for the valuable comments concerning our manuscript. We believe these suggestion/criticisms have helped us to improve our manuscript. We are glad to learn of the reviewer’s opinion that our manuscript may be suitable for publication in Healthcare after some revision. Please find attached a revised version of our manuscript entitled “Proteomics Analysis of Urine to Examine Physical Effects of Warm Nano Mist Sauna Bathing” by Hirao et al. (healthcare-496639). The manuscript has been revised in accordance with the reviewer’s comment and all the issues raised have been addressed. In Changes to the manuscript are highlighted in red and one of the table is added.

              We greatly appreciate the careful assessment of our manuscript by yourself and the reviewers. We hope the manuscript is now acceptable for publication in Healthcare.

Yours sincerely,

Tadashi Yamamoto, M.D & Ph.D.

Response to the comments

Comment-1

              In the introduction at the beginning of the first line there is a repeated letter.

Response-1

              We removed “T” from manuscript.

Comment-2

The non-nano mist sauna is sauna or is it sauna wet but not with nano particles?

Response-2

              The non-nano mist sauna is sauna wet but without nano particles.

Comment-3

              This is a case study, because the sample is very small, and because it was only made in men? women have no benefits, or is there any other reason?

Response-3

              This study is describing about the profiling of effect with or without nano-mist sauna. Furthermore, 40 samples were used for this analysis. The samples were obtained from only men. Because this nano-mist sauna system was supported from company and it was difficult to recruite women.

Comment-4

              How many times have you had a sauna for how long? The urine samples were collected before the test was started or were done every time before doing sauna ... if yes, was it done more than once !?

Response-4

              For the sample collection, there was a lack of explanation. The schedule of collecting urine is as follow; At 1st day, urine was collected for before taking sauna. Then volunteer took sauna every day for 2 weeks. At end of 14th days, we again collected the urine for after taking sauna. From your suggestion, we corrected the sentence “At 1st day, urine was collected for before taking sauna. Then volunteer took sauna every day for 2 weeks. At end of 14th days, we again collected the urine for after taking sauna.” at Page 2 lines 70-72.

Comment-5

              Table 1 is not explicit and should be improved so as to be clearer

Response-5

              Thank you for your suggestion. We changed the Table 1 to show more clear.

Comment-6

              The conclusion relates little to the results obtained with the benefits of the sauna, to say that the nano sauna has benefits because of humidity, temperature etc. without explanation seems to me quite vague and poor.

Response-6

              Thank you for your suggestion. We chebged Conclusion section more detail.

Round  2

Reviewer 1 Report

Hirao et al have done a good job of responding to comments and have an improved manuscript as a result. There are just a few comments that need to be addressed.

Page 2 line 72: change to “On the first day urine was collected prior to starting the study, a second urine sample was taken following 2 weeks of daily sauna use.”

How was the aforementioned urine sample taken? Was it a 24hr collections or a spot urine collection? If the later was it taken at the same time on both occasions?

Page 3 line 90: Change to “The digestion was stopped by the addition of 1 uL of 50%.....

Figure 3: The error bars are obscuring the number of proteins identified

Table 2: This is still missing aspects from MIAPE such as the number of peptide sequences, % protein coverage e.t.c. http://www.psidev.info/sites/default/files/2018-03/MIAPE_MSI_1.1.pdf

Page 7 lines 200-204: If a representative and informative normal urinary protein database is required then provide the information of all the proteins identified in this study in the supplementary information. This is a standard practice in proteomics.

Author Response

Dear Reviewer 1

              We would like to thank you for the valuable comments concerning our manuscript on 2nd revise. We believe these suggestion/criticisms have helped us to improve our manuscript. We are glad to learn of the reviewer’s opinion that our manuscript may be suitable for publication in Healthcare after this revision. Please find attached a revised version of our manuscript entitled “Proteomics Analysis of Urine to Examine Physical Effects of Warm Nano Mist Sauna Bathing” by Hirao et al. (healthcare-496639). The manuscript has been revised in accordance with the reviewer’s comment and all the issues raised have been addressed.

              We greatly appreciate the careful assessment of our manuscript by yourself and the reviewers. We hope the manuscript is now acceptable for publication in Healthcare.

Yours sincerely,

Tadashi Yamamoto

Response to the comments

I appreciate your deep reading and well understanding in our manuscript. Your comments are totally just in the focus of our study, and its are great help to improve our study so much. I would like to reply your comments.

Comment-1

              Page 2 line 72: change to “On the first day urine was collected prior to starting the study, a second urine sample was taken following 2 weeks of daily sauna use.”

Response-1

              Thnak you for your kind fixing of our English. We followed your suggestion. We highlighted in yellow on manuscript.

Comment-2

              How was the aforementioned urine sample taken? Was it a 24hr collections or a spot urine collection? If the later was it taken at the same time on both occasions?

Response-2

              The urine sample was collected at a spot. In this experiment, we adjusted as posible as same occasion at every urine sample collection.

Comment-3

              Page 3 line 90: Change to “The digestion was stopped by the addition of 1 uL of 50%.....

Response-3

              Again thank you for your editing. We followed your suggestion. We highlighted in yellow on manuscript.

Comment-4

              Figure 3: The error bars are obscuring the number of proteins identified

Response-4

              We agree with your point. We modified the Figure 3 more clearly to see.

Comment-5

              Table 2: This is still missing aspects from MIAPE such as the number of peptide sequences, % protein coverage e.t.c. http://www.psidev.info/sites/default/files/2018-03/MIAPE_MSI_1.1.pdf

Response-5

              We are sorry for misunderstaning of your suggestion. We checked more precisely the MIAPE guideline.

Table 2 is just want to show the protein list, which related to the lysosome pathway at more than 2 times fold changed. Each proteins were pre-analyzed by SIN and then calculated. Actually, all the proteins which identified in this study has such kind of scores and coverage. However, these data were not used in this table.

Comment-6

              Page 7 lines 200-204: If a representative and informative normal urinary protein database is required then provide the information of all the proteins identified in this study in the supplementary information. This is a standard practice in proteomics.

Response-6

              We totaly agree at this point. We provide the information of all the proteins and SIN value identified in this study. Also, we already uploaded all the raw data and MASCOT search result at JPOST repository (http://jpostdb.org/) for more detail of each parameter.

Reviewer 2 Report

Thank you for following the suggestions, it seems clearer to me now, although I think it should be a more universal study, doing the study only in men greatly impoverishes the result.
On conclusion there is an orthographic error.

Author Response

Dear Reviewer 2

              We would like to thank you for the valuable comments concerning our manuscript on 2nd revise. We believe these suggestion/criticisms have helped us to improve our manuscript. We are glad to learn of the reviewer’s opinion that our manuscript may be suitable for publication in Healthcare after this revision. Please find attached a revised version of our manuscript entitled “Proteomics Analysis of Urine to Examine Physical Effects of Warm Nano Mist Sauna Bathing” by Hirao et al. (healthcare-496639). The manuscript has been revised in accordance with the reviewer’s comment and all the issues raised have been addressed.

              We greatly appreciate the careful assessment of our manuscript by yourself and the reviewers. We hope the manuscript is now acceptable for publication in Healthcare.

Yours sincerely,

Tadashi Yamamoto

Response to the comments

We appreciate your deep reading and well understanding in our manuscript. Your comments are totally just in the focus of our study, and its are great help to improve our study so much.

About samples of women, we will think about it for further analysis. We also fixed english on conclusion. We thank you so much for your reviewing.